# A Closer Look at Machine Learning Effectiveness in Android Malware Detection

**Filippos Giannakas** *[ID], **Vasileios Kouliaridis** †[ID] and **Georgios Kambourakis** †[ID]

Department of Information and Communication Engineering, University of the Aegean, 83200 Karlovasi, Samos, Greece
* Correspondence: fgiannakas@aegean.gr
† These authors contributed equally to this work.

**Abstract:** Nowadays, with the increasing usage of Android devices in daily life activities, malware has been increasing rapidly, putting peoples' security and privacy at risk. To mitigate this threat, several researchers have proposed different methods to detect Android malware. Recently, machine learning based models have been explored by a significant mass of researchers checking for Android malware. However, selecting the most appropriate model is not straightforward, since there are several aspects that must be considered. Contributing to this domain, the current paper explores Android malware detection from diverse perspectives; this is achieved by optimizing and evaluating various machine learning algorithms. Specifically, we conducted an experiment for training, optimizing, and evaluating 27 machine learning algorithms, and a Deep Neural Network (DNN). During the optimization phase, we performed hyperparameter analysis using the Optuna framework. The evaluation phase includes the measurement of different performance metrics against a contemporary, rich dataset, to conclude with the most accurate model. The best model was further interpreted by conducting feature analysis, using the Shapley Additive Explanations (SHAP) framework. Our experiment results showed that the best model is the DNN consisting of four layers (two hidden), using the Adamax optimizer, as well as the Binary Cross-Entropy (loss), and the Softsign activation functions. The model succeeded with 86% prediction accuracy, while the balanced accuracy, the F1-score, and the ROC-AUC metrics were at 82%.

**Keywords:** malware detection; security; android; machine learning; neural network; deep learning; optimization; feature importance

## 1. Introduction

With the growing development of Android applications and the plethora of services offered by mobile devices, security threats are on the rise. This trend is further exacerbated in the context of ongoing crises, including the coronavirus one, which brought along a sudden need for businesses and their employees to start or increase working from home. Indeed, as an example, a recent report from McAfee [1] states that cyber criminals have been exploiting the global interest in the COVID-19 pandemic and vaccines by creating fake applications masquerading as official health department mobile applications.

In the mobile arena, malware is becoming the most prominent and dangerous threat that causes various security incidents, and result in a range of financial damage. Especially for devices based on the Android platform, each year, new and more sophisticated malware is being detected [1]. In this respect, contemporary malware detection techniques, including heuristic detection and signature-based detection, are not considered sufficient anymore to detect new malicious applications [2].

In an effort to detect and confront Android malware, various approaches have been proposed so far. In general, Android malware detection can be categorized into signature-based and anomaly-based. The latter usually employs Machine Learning (ML) to dis-

tinguish anomalies, i.e., deviations from the trained model, which are regarded as malicious behavior.

Furthermore, malware detection schemes can be classified into two main categories: static analysis and dynamic analysis. Static analysis uses syntactic features that can be extracted from each Android Package Kit (APK) file, without executing the application. This is considered a much safer and quicker process that also reduces the overall computational overhead. On the other hand, dynamic analysis requires the application to be executed on either a real device or a simulated environment, e.g., a Virtual Machine (VM). This is an important factor which makes research on this field lean towards static analysis [3].

In this context, in order to handle the detection of sophisticated, modern mobile malware, as well as the demand for more accurate predictions, diverse techniques such as ML algorithms are brought to the foreground [3,4]. Specifically, ML is a sub-set of Artificial Intelligence (AI) that gains applicability in various domains, while different mobile applications were developed. Especially in security targeting mobile devices, legacy ML algorithms such as Support Vector Machine (SVM), Logistic Regression (LR), and Decision Tree (DT) have been extensively assessed, showing promising results in classifying Android malware. Overall, ML models can be used to overcome the limitations of traditional detection methods and provide superior prediction scores [5].

More recently, several researchers have started to employ different Neural Networks (NN) for anomaly-based malware detection. Specifically, Deep Neural Networks (DNN), Convolutional Neural Networks (CNN), Recurrent Neural Networks (RNN), and Feed-Forward networks (FFN) have been applied to various detection schemes with promising results.

However, applying ML techniques for predicting malware applications is generally a cumbersome process, due to various factors that influence their training and prediction accuracy, and therefore such methods call for careful and thorough exploration. The first major consideration is the quality and the quantity of the input data used for training the ML algorithms. Furthermore, it is also crucial to inspect the various ML algorithms for comparing their detection performance, which in turn will assist in finding the more suitable one for a given problem. Finally, ML hyperparameter tuning is also key to the prediction performance and may include the enabling of techniques, including early stopping classifiers, the use of different topologies and activation functions, as well as the selection of different batches and epochs [6].

Altogether, the current work intends to investigate in a more comprehensive manner the use of ML algorithms for mobile malware detection, by applying different optimization techniques. Specifically, we explored and validated the performance of 28 different supervised and semi-supervised ML algorithms, including a DNN model, regarding their capacity in identifying Android malware. We conducted a comparative analysis in terms of prediction accuracy, and other relevant key metrics. We proceeded with a hyperparameter tuning of the selected DNN model, by using the Optuna framework, for exploring further its prediction accuracy. Last but not least, due to the existence of different input data, a side goal of the current study is to shed light on these input features that significantly affect the performance of malware prediction. Precisely, we enabled the SHAP framework for scrutinizing the ML algorithms and revealed key classification features that affect prediction performance.

To summarize, the work at hand contributes to the following goals:

- A large dataset of contemporary malware is collected to extract features using static analysis;
- Twenty seven different ML models were trained, using the aforementioned dataset in an effort to find the best performer;
- A DNN model is tuned and optimized after conducting hyperparameters importance analysis, by using the Optuna framework on our benchmark dataset.
- Feature importance analysis is performed using the SHAP framework on the best performing ML model to reveal the most significant classification features.

The outline of the rest of the paper is as follows: The next section focuses on the existing work on the field. Section 3 details the methodology used to conduct this study, including the utilized dataset, the testbed, and the relevant performance metrics. Section 4 focuses on the evaluation of 27 shallow ML models, while Section 5 concentrates on DNN model evaluation. Section 6 elaborates on feature importance. The final section concludes and provides future directions.

## 2. Related Work

As of today, several works in the literature relied on Deep Learning (DL) for malware detection in the Android platform. This section offers a chronologically ordered review of recent works on this topic. Specifically, we concentrate on studies published over the last five years, that is, from 2018 to 2022, considering only contributions that employed static analysis for feature extraction and DL for classification. Table 1 compares our approach with the works included in this section based on five criteria, namely number of models examined and compared, optimization techniques, hyperparameter tuning, feature importance, and dataset(s) used to train the models. For the latter criterion, we consider a dataset as contemporary if it comprises malware samples not older than five years. A plus symbol designates that this subject is addressed by the respective work, while a hyphen denotes the opposite.

Dongfang et al. [7] proposed a DL-based method for Android malware detection. The authors employed static analysis to extract features, i.e., permissions and API calls from the Drebin dataset [8] and used them to train their models. The authors reported an accuracy of 97.16%.

Karbab et al. [9] contributed an automatic Android malware detection framework called "MalDozer". The latter can extract API call sequences from Android applications and detect malicious patterns using sequences of classification by employing DL techniques. The authors evaluated their scheme against a large dataset comprising Android applications from three well-known datasets, namely, MalGenome [10], Drebin [8], as well as their own dataset called MalDozer. Their results yielded an F1 score between 96% and 99%.

Wenjia et al. [11] proposed a DL-based scheme for Android malware characterization and identification. The authors extracted various features from the Drebin dataset [8], i.e., permissions, intents, IP addresses and URLs, and API calls. Different weights were given to classification feature combinations. Based on previous work, these weight-adjusted features were then used to train their model. The experimental results reported an accuracy of over 90% using 237 features.

Xu et al. [12] presented a DL-based Android malware detection technique, which leverages both XML files and bytecode of Android applications. Firstly, DeepRefiner retrieves XML values by performing lightweight preprocessing on all XML files included in the application, to extract information about the required resources. If an application is considered suspicious, it is further analyzed by looking at the bytecode semantics, which provides comprehensive information about programming behaviors. The authors evaluated the detection performance of DeepRefiner over 62,915 malicious and 47,525 benign applications collected from the VirusShare [13] and MassVet datasets and reported an accuracy of 97.74%.

Zegzhda et al. [14] proposed a DL-based approach for Android malware detection, which uses a CNN to identify malicious Android applications. The authors extracted API calls from a dataset comprising 7214 benign and 24,439 malicious samples. The benign applications were collected from third-party libraries and verified using VirusTotal [15], while malicious samples were collected from the Android Malware Dataset (AMD) [16]. The authors evaluated their results and reported an Accuracy of 93.64%.

Zhiwu et al. [17] proposed CDGDroid, an approach for Android malware detection based on DL. Their approach relies on the semantics graph representations, i.e., control flow graph, data flow graph, and their possible combinations, as the features to characterize an Android application as malware or benign. These graphs are then encoded into matrices,

which are used to train the classification model through CNN. The authors conducted experiments using various datasets, namely Marvin [18], Drebin [8], VirusShare [13], and ContagioDump, and reported an F1 score of up to 98.72%.

Kim et al. [19] presented a framework that uses a multimodal DL method to detect Android malware applications. The proposed framework uses seven types of features stemming from static analysis, namely strings, method opcodes, API calls, shared library function opcodes, permissions, components, and environment settings. To evaluate the performance of their framework, the authors collected 41,260 applications, out of which 20K were benign. Their results showed a precision, recall, F1, and Accuracy rate of 0.98, 0.99, 0.99, and 0.98, respectively.

Masum and Shahriar [20] proposed a DL-based framework for Android malware classification, called "Droid-NNet". Droid-NNet's Neural Network consists of three layers, namely input, hidden, and output. A threshold is applied to the output layer to classify the examined application as malware or benign. The input layer comprises 215 neurons, which is the number of features used during the training phase. The hidden layer contains 25 neurons, while the output layer includes only one neuron since the classification is binary. Additionally, the authors applied binary cross-entropy as a loss function and an Adaptive Moment Estimation (Adam) optimizer for calculating error and updating the relevant parameters. To train the model, the authors used the 215 static features provided by the Drebin [8] and Malgenome [10] datasets. Their results yielded an F-beta rate of 0.992 and 0.988, on Malgenome and Drebin datasets, respectively.

Niu et al. [21] presented a DL-based approach for Android malware detection based on OpCode-level Function Call Graph (FCG). The FCG was obtained through static analysis of Operation Code (OpCode). The authors used the Long Short-Term Memory (LSTM) model and trained it using 1796 Android malware samples collected from the Virusshare [13] and AndroZoo [22] datasets, as well as 1K benign Android applications. Their experimental results showed that their approach was able to achieve an accuracy of 97%.

Pektas and Acarman [23] contributed a malware detection method that relies on a pseudo-dynamic analysis of Android applications and constructs an API call graph for each execution path. In other words, the proposed approach focuses on information extraction related to an application's execution paths and embedding of graphs into a low-dimensional feature vector, which is used to train a DNN. According to the authors, the trained DNN is able to detect API call graph and binary code similarities to determine whether an application is malicious or not. Finally, the DL parameters are tuned, and the Tree-structured Parzen Estimator is applied to seek the optimum parameters in the parameter hyper-plane. Their method achieved an F1 and accuracy score of 98.65% and 98.86%, respectively, on the AMD dataset [16].

Zou et al. [24] proposed ByteDroid, an Android malware detection scheme that analyzes Dalvik bytecode using DL. ByteDroid resizes the raw bytecode and constructs a learnable vector representation as the input to the neural network. Next, ByteDroid adopts a CNN to automatically extract the malware features and perform the classification. The authors tested their method against four datasets, namely FalDroid [25], PRAGuard [26], Virushare [13], and a dataset from Kang et al. [27], and reported a detection rate of 92.17%.

Karbab et al. [28] proposed a DL-based Android malware detection framework called "PetaDroid". This framework analyzes bytecode and an ensemble of CNN to detect Android malware. First, for each sample, the PetaDroid disassembles the DEX bytecode into Dalvik assembly to create sequences of canonical instructions. Additionally, the framework utilizes code-fragment randomization during the training phase to render the model more resilient to common obfuscation techniques. The authors evaluated PetaDroid against various datasets, namely MalGenome, Drebin, MalDozer, AMD, and VirusShare. PetaDroid achieved an F1 score of 98% to 99%, under different evaluation settings with high homogeneity in the produced clusters (96%).

Millar et al. [29] contributed a DL-based Android malware detector with a CNN-based approach for analyzing API call sequences. Their approach employs static analysis to

extract opcodes, permissions, and API calls from each Android application. The authors carried out various experiments, including hyper-parameter tuning for the opcodes CNN and the APIs CNN and zero-day malware detection. The proposed model achieved an F1 score of 99.2% and 99.6% using the Drebin and AMD datasets, respectively. Additionally, the authors reported an 81% and 91% detection rate during the zero-day experiments on the AMD and Drebin datasets, respectively.

Vu et al. [30] contributed an approach that trains a CNN to classify mobile malware. In addition, their approach converts an application's source code, i.e., API calls extracted from APK files, into a two-dimensional adjacency matrix, to improve classification performance. According to the authors, their approach allows better feature embedding than when using feature vectors, and can achieve comparable performance to call-graph analysis. The authors trained their model using samples from the Drebin and AMD datasets, and achieved a detection and classification rate of 98.26% and over 97%, respectively.

Zhang et al. [31] proposed "TC-Droid", an automatic Android malware detection framework that employs text classification. Precisely, TC-Droid feeds on the text sequence of APIs analysis reports, generated by AndroPyTool and uses CNN to gather significant information. Specifically, TC-Droid analyzes four types of static features, i.e., permissions, services, intents, and receivers. The authors evaluated their framework using malware samples from ContagioDump and MalGenome datasets and reported an accuracy of 96.6%.

Yumlembam et al. [32] proposed the use of a Graph Neural Networks (GNN) based classifier to generate an API graph embedding fed by permissions and intents to train multiple ML and DL algorithms for Android malware detection. Their approach achieved an accuracy of 98.33% and 98.68% with the CICMaldroid2020 [33] and Drebin datasets, respectively.

Musikawan et al. [34] introduced a DL-based Android malware classifier, in which the predictive output of each of the hidden layers given by a base classifier is combined via a meta-classifier to produce the final prediction. The authors tested their approach on two dynamic and one static datasets. On the static dataset, i.e., CICMalDroid2020 [33], their approach achieved an F1 score of 98.1%.

**Table 1.** Overview of the related work. A plus sign denotes that the respective work addresses the criterion of the corresponding column. The figures in the third column denote the number of ML models examined by each study. A dash in the same column means that the respective work did not make a comparison between two or more different models.

| Work | Year | Models Compared | Model Optimization | Hyperparameters Tuning | Feature Importance | Contemporary Dataset |
|------|------|-----------------|--------------------|-----------------------|--------------------|----------------------|
| [7] | 2018 | - | - | - | - | - |
| [9] | 2018 | - | - | - | - | - |
| [11] | 2018 | - | - | - | - | - |
| [12] | 2018 | - | - | - | - | - |
| [14] | 2018 | - | - | - | - | - |
| [17] | 2018 | - | - | - | - | - |
| [19] | 2019 | - | - | - | - | - |
| [20] | 2020 | - | - | - | - | - |
| [21] | 2020 | 8 | - | - | - | + |
| [23] | 2020 | - | - | + | - | - |
| [24] | 2020 | - | - | + | - | - |
| [28] | 2021 | - | - | + | - | - |
| [29] | 2021 | - | - | + | - | - |
| [30] | 2021 | - | - | - | + | - |
| [31] | 2021 | - | - | - | - | - |
| [32] | 2022 | 12 | + | + | + | + |
| [34] | 2022 | - | + | + | + | + |
| This work | 2022 | 27 | + | + | + | + |

As observed from Table 1, the relevant recent work on this topic falls short of providing an adequate, full-fledged view of Android malware detection through ML techniques. That is, from the third column of the Table, it is obvious that the great majority of works rely only on a single ML model. In this respect, the current study sees the problem from multiple viewpoints, namely by examining and comparing the detection performance of diverse ML models, both shallow and DL. On top of that, the provided analysis includes hyperparameter tuning and feature importance evaluation. In this regard, the methodology and outcomes of this work can serve as a guide and reference point for future research in this rapidly evolving field.

## 3. Methodology

### 3.1. Dataset

For the needs of the current work, 1000 malware samples were collected from the most contemporary benchmark dataset, namely AndroZoo [22], dated from 2017 to 2020. AndroZoo is a well-known and widely used collection of Android applications gathered from various sources, including the official Google Play application market [35]. It includes new and more sophisticated malware samples in comparison to older datasets, e.g., Drebin [8]. Each of the chosen applications was cross-examined by several antivirus products. The (balanced) dataset also contained 1000 benign applications, collected from Google Play. The latter applications were also dated from 2017 to 2020. The list of malicious and benign samples (applications) used in the experiments can be found in a publicly accessible GitHub repository, as given in the "Data Availability Statement" section.

### 3.2. Data Analysis

Static analysis was performed on all the samples using the open-source tool Androtomist [36], which is able to extract various features from an Android application by decompiling its APK file. This work relies on two feature categories, namely permissions and intents. This is carried out because the aforementioned feature categories are the commonest in the relevant research [3], and therefore it allows for easy comparison with previous or future studies. Precisely, the tool extracted the aforementioned features from each application's *Manifest.xml* file to create a feature vector, i.e., a binary representation of each distinct feature.

### 3.3. Research Design and Testbed

In the conducted experiment detailed further down in Section 4, Android malware detection is shaped as a binary classification problem, for which either an ML or a DNN model can be utilized. However, choosing the most accurate ML model for predicting whether or not an Android application is indeed malware is considered a cumbersome procedure. This is due to the existence of various parameters and biases that influence the final prediction. These include the different possible ML/DNN models, the input data, the existence of various activation and loss functions, as well as the many hyperparameters that the different ML/DNN models enable. All the above issues compose a complex setting that needs to be carefully explored before recommending the most suitable and accurate model.

In this context, the experiments carried out in the context of the present work were split into two distinct phases. The main purpose of the first phase is to test and evaluate a variety of ML algorithms, including a DNN model, for identifying the most suitable and accurate one. In the second phase, the model that achieved the best score in terms of prediction was re-trained and re-evaluated using other quality metrics. In this way, one can draw more accurate conclusions about its prediction performance. During the above-mentioned phases, GPU support was also enabled for accelerating both the training and evaluating processes of the ML/DNN models.

The selection of the ML algorithms was based on two criteria. First, its prevalence in similar works, and second, the implementation of each selected algorithm to be readily available in well-known ML libraries. Thus, shallow classification on the dataset was

conducted against 27 ML models as shown in the first column of Table 2. DNN classification on the other hand was based on different structures, as well as on specific optimizers, activation, and loss functions. Specifically, for implementing and evaluating both shallow and DNN models, we relied on the NumPy v.1.19.5, scikit-learn v.0.23.1, SHAP v.0.40.0, TensorFlow v.2.4.0, Keras v.2.4.0, and Optuna v.2.10.0 in Python v3.7.0. Moreover, the training process was offloaded to a GPU using the NVIDIA GPU v.526.86, CUDA API v.12.0, and CuDNN SDK 7.6. The implementation of the above mentioned models, as well as the conducted experiments, were carried out on an MS Windows 10 Pro machine, incorporating an Intel Core i7-7700HQ processor, 16 GB DDR4-2400 RAM, and an NVIDIA GeForce GTX 1050 Mobile with 4 GB RAM GPU.

**Table 2.** Evaluation results per shallow ML algorithm. The classifiers are ordered based on the accuracy metric.

| Algorithm | Prediction Acc. | Balanced Acc. | F1 Score | ROC-AUC |
|---|---|---|---|---|
| XGBClassifier | 0.84 | 0.84 | 0.84 | 0.84 |
| LGBMClassifier | 0.82 | 0.82 | 0.82 | 0.82 |
| BaggingClassifier | 0.82 | 0.82 | 0.82 | 0.82 |
| ExtraTreesClassifier | 0.81 | 0.80 | 0.80 | 0.81 |
| DecisionTreeClassifier | 0.80 | 0.80 | 0.80 | 0.80 |
| RandomForestClassifier | 0.79 | 0.79 | 0.79 | 0.79 |
| AdaBoostClassifier | 0.78 | 0.78 | 0.78 | 0.78 |
| KNeighborsClassifier | 0.76 | 0.75 | 0.75 | 0.76 |
| LinearDiscriminantAnalysis | 0.76 | 0.75 | 0.75 | 0.76 |
| RidgeClassifier | 0.76 | 0.75 | 0.75 | 0.76 |
| LinearSVC | 0.76 | 0.75 | 0.75 | 0.76 |
| RidgeClassifierCV | 0.76 | 0.75 | 0.75 | 0.75 |
| LogisticRegression | 0.76 | 0.75 | 0.75 | 0.75 |
| SGDClassifier | 0.76 | 0.75 | 0.75 | 0.75 |
| CalibratedClassifierCV | 0.75 | 0.74 | 0.74 | 0.75 |
| ExtraTreeClassifier | 0.74 | 0.74 | 0.74 | 0.75 |
| NuSVC | 0.74 | 0.73 | 0.73 | 0.74 |
| SVC | 0.73 | 0.73 | 0.73 | 0.73 |
| Perceptron | 0.72 | 0.71 | 0.71 | 0.71 |
| QuadraticDiscriminantAnalysis | 0.72 | 0.71 | 0.71 | 0.71 |
| PassiveAggressiveClassifier | 0.71 | 0.70 | 0.70 | 0.70 |
| GaussianNB | 0.72 | 0.70 | 0.70 | 0.70 |
| NearestCentroid | 0.62 | 0.63 | 0.63 | 0.62 |
| LabelSpreading | 0.65 | 0.62 | 0.62 | 0.61 |
| LabelPropagation | 0.65 | 0.62 | 0.62 | 0.61 |
| BernoulliNB | 0.61 | 0.62 | 0.62 | 0.62 |
| DummyClassifier | 0.47 | 0.47 | 0.47 | 0.47 |

### 3.4. Performance Metrics

In the literature, various performance metrics exist for evaluating the accuracy of an ML model. For the purposes of the current research, the following metrics were considered: Prediction Accuracy, Balanced Accuracy, F1 score, and Area Under the Curve-Receiver

Operating Characteristics (AUC-ROC). The latter two metrics are key for assessing the preciseness and robustness of the model to produce superior predictions. If the dataset is balanced (as in our case), then accuracy is a valid metric [36]. Otherwise, F1 should be the prominent metric.

Specifically, the prediction accuracy of the model is calculated by Equation (1), which denotes the accuracy of the model to make correct predictions:

$$Prediction\,Accuracy = \frac{\text{Number of correct predictions}}{\text{Total number of predictions}} \times 100 \tag{1}$$

The F1 score metric is the harmonic mean of precision and recall metrics, which are discussed in detail below in this subsection. This metric is computed by Equation (2). The maximum result value for this metric is 1, meaning that both the precision and recall are perfect, while the minimum is 0:

$$F1 = \frac{2}{\frac{1}{precision} + \frac{1}{recall}} = \frac{2 * precision * recall}{precision + recall} \tag{2}$$

The accuracy of the model is also measured by the AUC-ROC curve. This metric shows how capable the model is of making correct predictions between the distinct classes. Precisely, the ROC is the probability curve, while the AUC represents the degree of separability. The higher the AUC, the better the model predicts the classes correctly. For computing the AUC-ROC metric, apart from the necessity to calculate the recall metric, one also needs to compute both the so-called Specificity (shown in Equation (3)), as well as the False Positive Rate (FPR), given in Equation (4). Last but not least, for observing any data imbalance between the two classes (malware or not), meaning that one class of the two appears more often than the other in the dataset, we also calculated the Balanced Accuracy:

$$\text{Specificity} = \frac{TN}{TN + FP} \tag{3}$$

$$\text{FPR} = 1 - \text{Specificity} = \frac{FP}{TN + FP} \tag{4}$$

## 4. Shallow Classifiers

### 4.1. Initial Analysis

The current section includes a large-scale evaluation of 27 different shallow ML algorithms against the dataset. At first, all the classifiers were trained and evaluated by keeping unchanged their default hyperparameter settings. For each algorithm, we extracted the relevant performance metrics as discussed previously in Section 3.4.

As shown in Table 2, the XGBoost classifier succeeded the best prediction results. Specifically, its prediction accuracy, balanced accuracy, F1-score, and ROC-AUC metrics were found to be 82%. Furthermore, for the same classifier, we calculated the log loss (negative log-likelihood), the classification error, and the confusion matrix. The latter is a 2D table that visualizes the performance of the model by illustrating the actual label values in columns, and the predicted labels in rows. These values indicate the True Negative (TN), True Positive (TP), False Positive (FP), and False Negative (FN) predictions of the model.

In more detail, the TP index shows the number of the correct predictions of the model, meaning that these applications are correctly identified as being malware by the model, whereas the FP indicates the applications that are falsely identified as being malware. In the same way, the TN value corresponds to the number of applications that are correctly predicted as goodware, whereas the FN shows the incorrect predictions of the applications as not being malware.

Figure 1 depicts the log loss (negative log-likelihood) and the classification error during the training and testing phase, as well as the ROC-AUC and the confusion matrix of the XGBoost classifier.

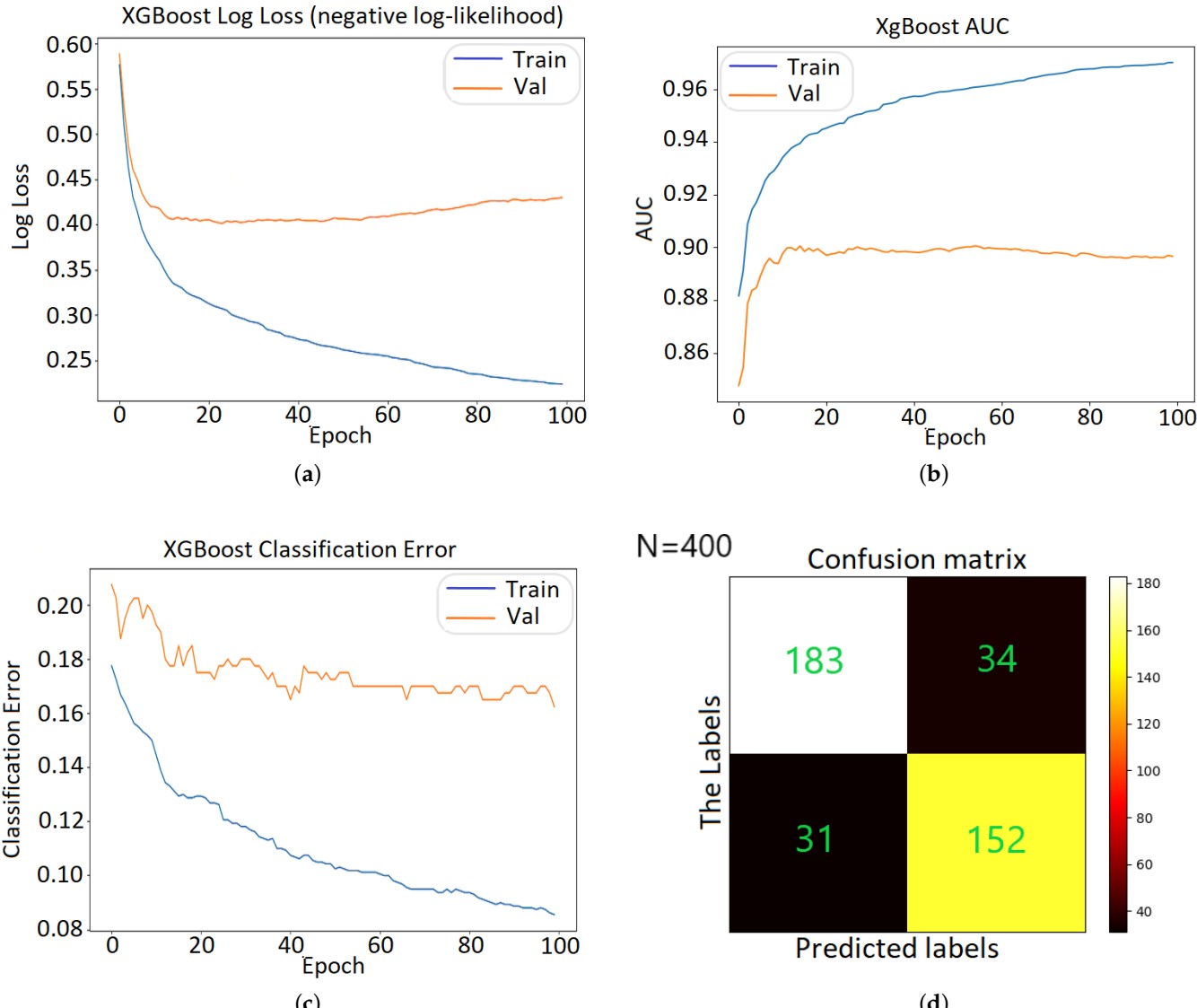

**Figure 1.** Performance curves of the XGBoost classifier: (**a**) log loss (negative log-likelihood); (**b**) AUC; (**c**) classification error; (**d**) confusion matrix.

Finally, other related metrics to the confusion matrix were also calculated for assessing the recall and the precision of the XGBoost model. Both of the latter metrics are useful for measuring the relevance of the model to make accurate predictions. The recall metric, which is also known as TP rate or sensitivity, is calculated by Equation (5). This was found to be 83% for the XGBoost classifier and depicts the proportion of the actual Android malware (positives values) that were correctly predicted by the model. In other words, this is the fraction that quantifies the number of malware (positive class) predictions made out over all Android malware (positive examples) contained in the dataset.

On the other hand, the precision metric designates the portion of the predicted Android malware (positive classes) that were actually identified correctly by the model (Equation (6)). This metric actually quantifies the number of positive classes that correctly belong to the correct prediction class, and in our case was found to be 82%:

$$Recall = \frac{TP}{TP + FN} \tag{5}$$

$$Precision = \frac{TP}{TP + FP} \tag{6}$$

### 4.2. ML Optimization

In the previous section, the considered ML algorithms were trained and evaluated by keeping their hyperparameter settings unchanged. However, the authors in [37–39] argue about the importance of fine-tuning the ML hyperparameters for improving the prediction accuracy of an ML algorithm. Among others, this may be achieved by using diverse methods and techniques, such as applying statistical methods, or using different quantifying measures for applying forward selection for the most important hyperparameters on different datasets.

Under this mindset, the XGboost algorithm, succeeded in obtaining the best accuracy results according to Table 2, was further scrutinized by fine-tuning its hyperparameters in an effort to optimize and improve further its prediction performance. For this purpose, the "Optuna" open-source framework [40] was used.

By observing the abstract view of the process in Figure 2, the data extracted from the Android sample were split into the training and validation sets. First off, the model was trained with the training dataset and produced the training evaluation results. Next, the model was validated with the validating dataset and the relevant performance results were extracted. Then, a hyperparameter tuning phase started for altering the parameters accordingly, and the model was re-trained. The new prediction results were compared to the previous ones, and if the model succeeded in obtaining better results, the new hyperparameter settings were kept. The final output of this iterating process is the best hyperparameter settings of the model that yielded the most accurate prediction results.

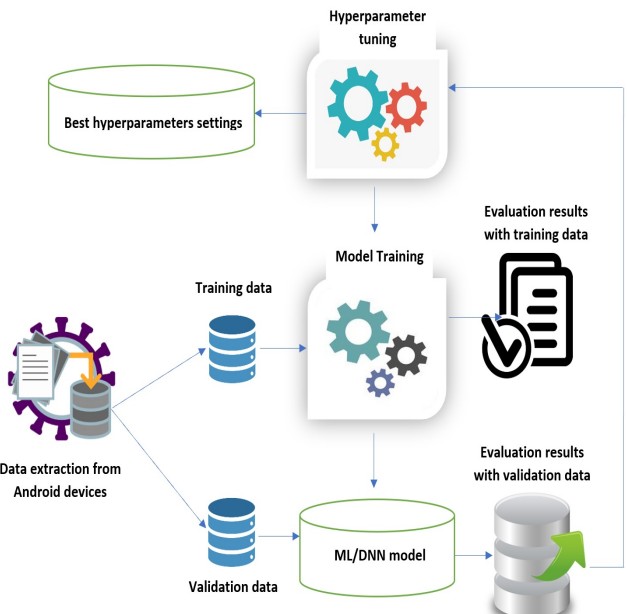

**Figure 2.** Abstract view of the training process and hyperparameter tuning of an ML model.

As shown in Table 3, after fine-tuning the XGBoost classifier, we extracted the optimized values for its hyperparameters. Based on these values, the classifier was reconfigured and both the training and evaluation phases were repeated. As observed from Table 3, the classifier improved its prediction accuracy by 2% and reached 86%. Furthermore, the F1-score and the ROC-AUC metrics were further improved and measured at 86.5% and 86.7%, respectively. The optimization history of the XGBoost classifier across the epochs is shown in Figure 3. In the figure, the blue dots denote the objective values across the different epochs, while the red line represents the best ones.

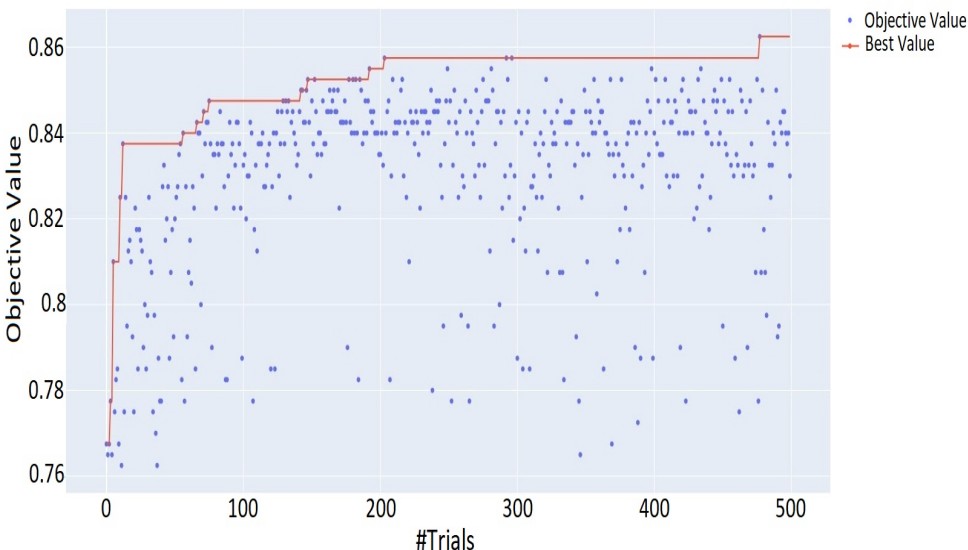

**Figure 3.** Optimization history of the XGboost classifier.

**Table 3.** XGboost's hyperparameters and its default and optimized values.

| Parameter Name | Description | Default Values | Optimized Values |
|---|---|---|---|
| n_estimators | Number of gradient boosted trees | 100 | 1700 |
| learning_rate | Boosting learning rate | 0.1 | 0.37 |
| reg_alpha | L1 regularization term on weights | 0 | 2 |
| reg_lambda | L2 regularization term on weights | 1 | 4 |
| gamma | Minimum loss reduction required to make a further partition on a leaf node of the tree | 0 | 0 |
| max_delta_step | Maximum delta step we allow each leaf output to be. If the value is set to 0, it means there is no constraint. If it is set to a positive value, it can help making the update step more conservative | 0 | 5 |
| max_depth | Maximum tree depth for base learners | 3 | 25 |
| colsample_bytree | Subsample ratio of columns when constructing each tree | 1 | 0.91 |
| colsample_bylevel | The subsample ratio of columns for each level | 1 | 0.78 |
| min_child_weight | Minimum sum of instance weight (hessian) needed in a child | 1 | 4 |
| n_iter_no_change | Number of repetitions without any change | 50 | 50 |

Additionally, Figure 4 depicts the percentage that each hyperparameter affects the objective value (prediction accuracy) of the XGboost classifier. Specifically, the default value of the "gamma" hyperparameter contributes to the optimal prediction accuracy of the model by 56%. In the same way, the default value of the colsample_bytree hyperparameter, after being altered to 0.91, contributes to the optimal prediction by 22%. No less important, the values of the hyperparameters reg_alpha, reg_lambda, and learning_rate improve the prediction accuracy of the classifier by 7%, 6%, and 6%, respectively, after changing correspondingly their default values to 2, 4, and 0.37. The rest of the hyperparameters contribute to the prediction results of the XGboost classifier less than 1%.

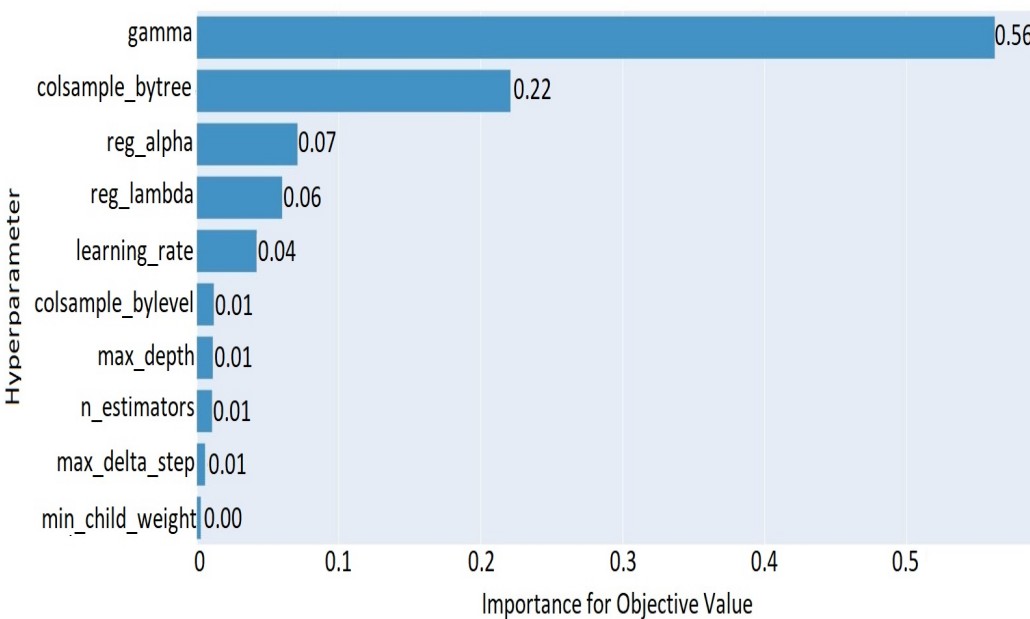

**Figure 4.** Hyperparameters' importance for the XGboost classifier.

The empirical distribution function (also known as the Empirical Cumulative Distribution Function) of the XGboost classifier is depicted in Figure 5. Specifically, it shows the cumulative probability across the objective and best value. Recall that generally the empirical distribution function describes a sample of observations of a given variable.

Last but not least, Figure 6 illustrates the relation between the multiple values of the hyperparameters in relation to the best value. Actually, this figure shows clearly the association of each hyperparameter value with the best score of the XGboost classifier.

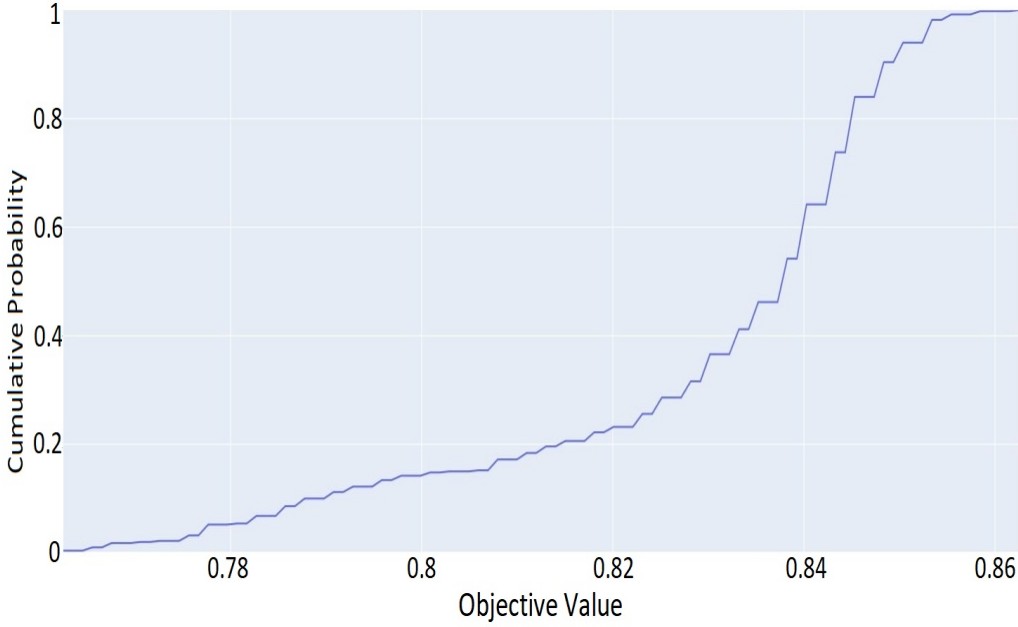

**Figure 5.** Cumulative probability distribution of the XGboost classifier.

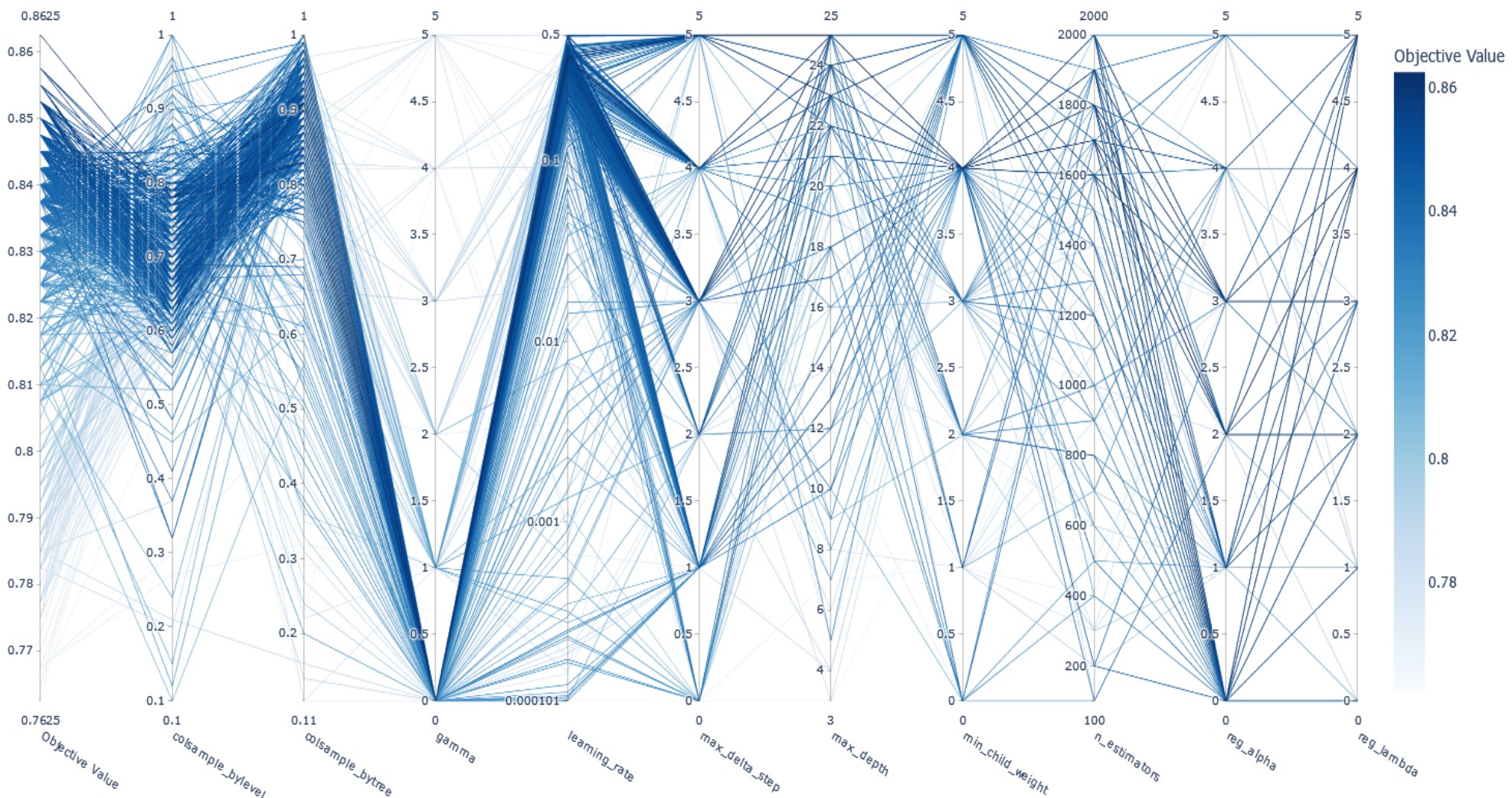

**Figure 6.** Hyperparameters' parallel coordination view for the XGboost classifier.

## 5. DNN Analysis

### 5.1. Preliminaries

There are several differences when comparing a ML model with a DNN. Precisely, some of them concern the training process of the DNN model and how it learns the way the weights of each input feature are assigned and automatically changed for improving the performance of the model, and so on. Furthermore, by comparing their structures, the DNN architecture is deeper and may comprise a number of hidden layers, in which various algorithms can be enabled for aiding surpassing some of the shortcomings of ML techniques. For example, a common problem for ML algorithms is the performance saturation, where the learning accuracy of the model reaches the maximum, meaning that it can not be improved further, even if one feeds the model with new input data.

Generally speaking, a DNN architecture that consists of L hidden layers uses Equation (7) to calculate the output for each layer. This is denoted by the $h^{(L)}(x)$ and indicates the output of layer L with arguments x:

$$f(x) = f([a^{(L+1)}(h^{(L)}(a^{(L)}(...(h^{(2)}(a^{(2)}(h(1)(a(x))))))))]) \tag{7}$$

Furthermore, during the learning phase, the final output is calculated by Equation (8). The a(x) input is a vector of arguments, and the result of the Equation (9) is also a vector. For example, $a^{(2)}$ denotes the argument of layer 2:

$$f_k(x) = f_k(a_k(x)) \tag{8}$$

$$a(x) = Wh(x) + b \tag{9}$$

As already pointed out, in the current research, Android malware detection is modeled as a binary classification problem. To find the most accurate DNN model, one should evaluate diverse architectures that include various numbers of layers and neurons and choose the most appropriate optimizer, activation, and loss function. Nevertheless, this is a laborious process due to the large number of combinations that need to be examined and taken into account during the learning phase. In the literature, several researchers choose to evaluate their models through a specific DNN structure and by enabling specific optimizers, activation, and loss functions. Therefore, this option may fail to conclude the most optimal and accurate model. Therefore, for making a conclusion about the best DNN model, we also applied the "Optuna" framework.

### 5.2. DNN Hyperparameter Optimization and Evaluation Results

The DNN hyperparameter tuning is considered an important, but not a straightforward task. This process is useful for making a conclusion about the most appropriate selection of an activation and optimization function, as well as for the best structure of the model. Additionally, one needs to enable an early stopping functionality, configure the batches-epochs, activate diagnostic procedures such as overfitting-underfitting during the learning process, etc.

The underfitting phenomenon appears in the NNs when either these have not been trained for enough time, or the training data are not significant enough to determine a meaningful relationship between the input and output variables. This phenomenon forces the prediction results of the neural networks to be poor. On the other hand, overfitting is a modeling error that occurs when the enabled function is too closely aligned to a limited set of data points. This means that the model is considered useful only to its initial dataset, and not to any other ones. Therefore, to avoid the overfitting problem, the dropout regularization [41] is applied to the NN, in order to skip some neurons' connections randomly, while the model is trained.

For this purpose, as pointed out previously, the Optuna open-source framework [40] was used again for optimizing the performance of the DNN model towards making more accurate Android malware predictions. Optuna created different structures of DNN models

and evaluated their performance prediction by altering their hyperparameters, as shown in Table 4. Additionally, we proceeded to further configurations of the model. That is, the DNN was tuned to use the dataset for 2000 times (epochs), and its internal parameters were set to be updated after a sample of 25 records (batches). Finally, Optuna was configured to run the optimization and evaluation process of the DNN model 50 times.

**Table 4.** DNN's hyperparameter configuration thresholds.

| Parameter Name | Description | Value(from) | Value(to) |
|---|---|---|---|
| n_layers | Number of layers | 1 | 6 |
| n_hidden | Number of hidden layers | 1 | 4 |
| learning_rate | Boosting learning rate | $e^{-5}$ | $e^{-1}$ |
| dropout | Dropout regularization to prevent overfitting | 0.2 | 0.5 |
| optimizer | Optimizer function | Adadelta, SGD, Adam, Adamax, Adagrad, Nadam, Nadam, Ftrl, RMSprop | |
| activationfunc | Activation function | sigmoid, softsign, elu, selu | |
| lossfunc | Loss function | binary_crossentropy, mean_squared_error | |

Figure 7 illustrates the way each hyperparameter affects the objective value (prediction accuracy) of the DNN model. As observed, the Dropout parameter at the input layer affects the optimal prediction accuracy (best value) of the model by 44%. Moreover, the learning rate (lr) parameter impacts the optimal prediction of the model by 21%. The number of units at the input layers and the number of layers contribute to the best value of the model by 20% and 15%, respectively. Another useful plot is also shown in Figure 8; it visualizes the relation between the multiple values of the hyperparameters and the best prediction value. Actually, this figure clarifies the association of the value for each hyperparameter towards the best value of the DNN model.

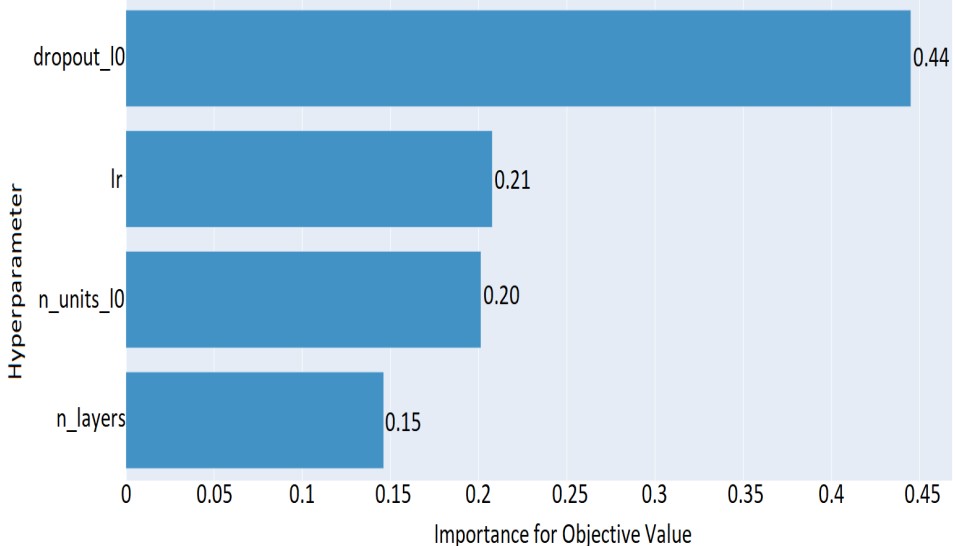

**Figure 7.** Hyperparameters' importance for the DNN model.

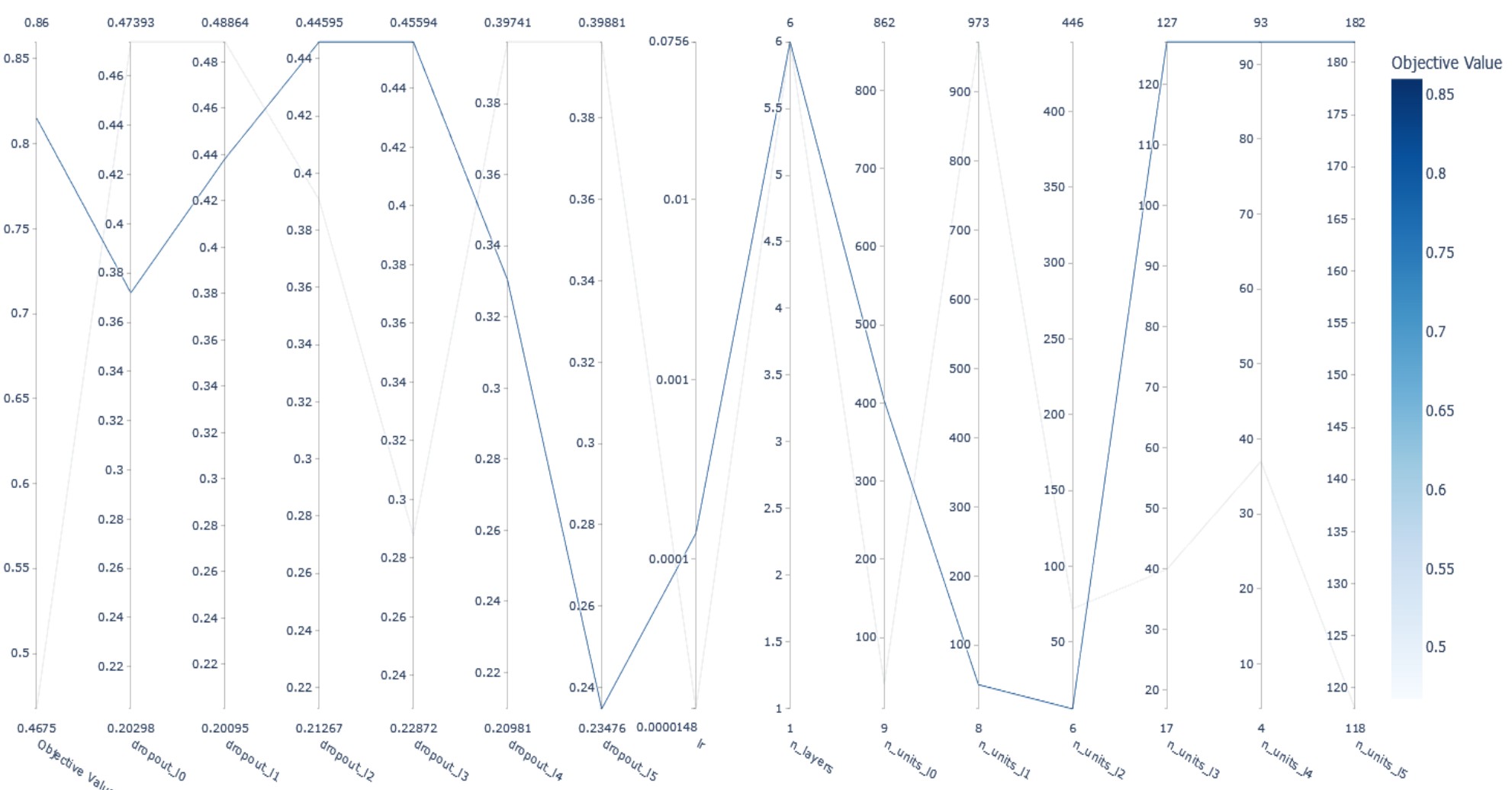

**Figure 8.** Hyperparameters' parallel coordination schema of the DNN model.

The empirical distribution function of the DNN model is shown in Figure 9. The aforementioned performance analysis suggests that the DNN model succeeded with higher performance results in predicting Android malware, compared to the shallow classifiers evaluated in Section 4.2.

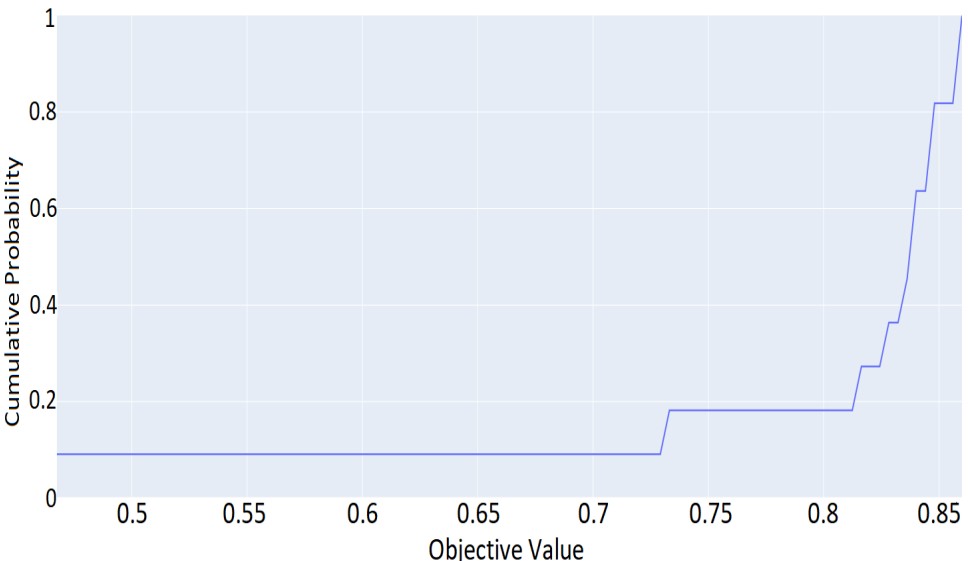

**Figure 9.** Cumulative probability distribution of the DNN model.

The rest of the performance metrics for the DNN model are illustrated in Figure 10. Specifically, the accuracy curves for the learning and prediction phase across the different epochs are shown in Figure 10a, while the cross-entropy (loss) is depicted in Figure 10b. Last but not least, the confusion matrix of the model is given in Figure 10c.

By observing the aforementioned evaluation results, the architecture of the DNN model with the highest accuracy prediction results consists of four layers, with two hidden ones. The abstract architecture view of the model is illustrated in Figure 11, whereas the parameters of each layer are recapitulated in Table 5. The optimized model enables the Adamax optimizer, as well as the Binary Cross-Entropy (loss) and the Softsign activation functions. The optimization history of the classifier across the epochs is shown in Figure 12. Specifically, the optimized prediction accuracy of the model reached 86%, which is better by 2% compared to the XGboost one.

**Table 5.** Total parameters of the DNN: 300,413. All the parameters are considered trainable.

| Layer (Type) | Output Shape | Params |
|---|---|---|
| dense (Dense) | (None, 238) | 238,952 |
| dense_1 (Dense) | (None, 91) | 21,749 |
| dense_2 (Dense) | (None, 427) | 39,284 |
| dense_3 (Dense) | (None, 1) | 428 |

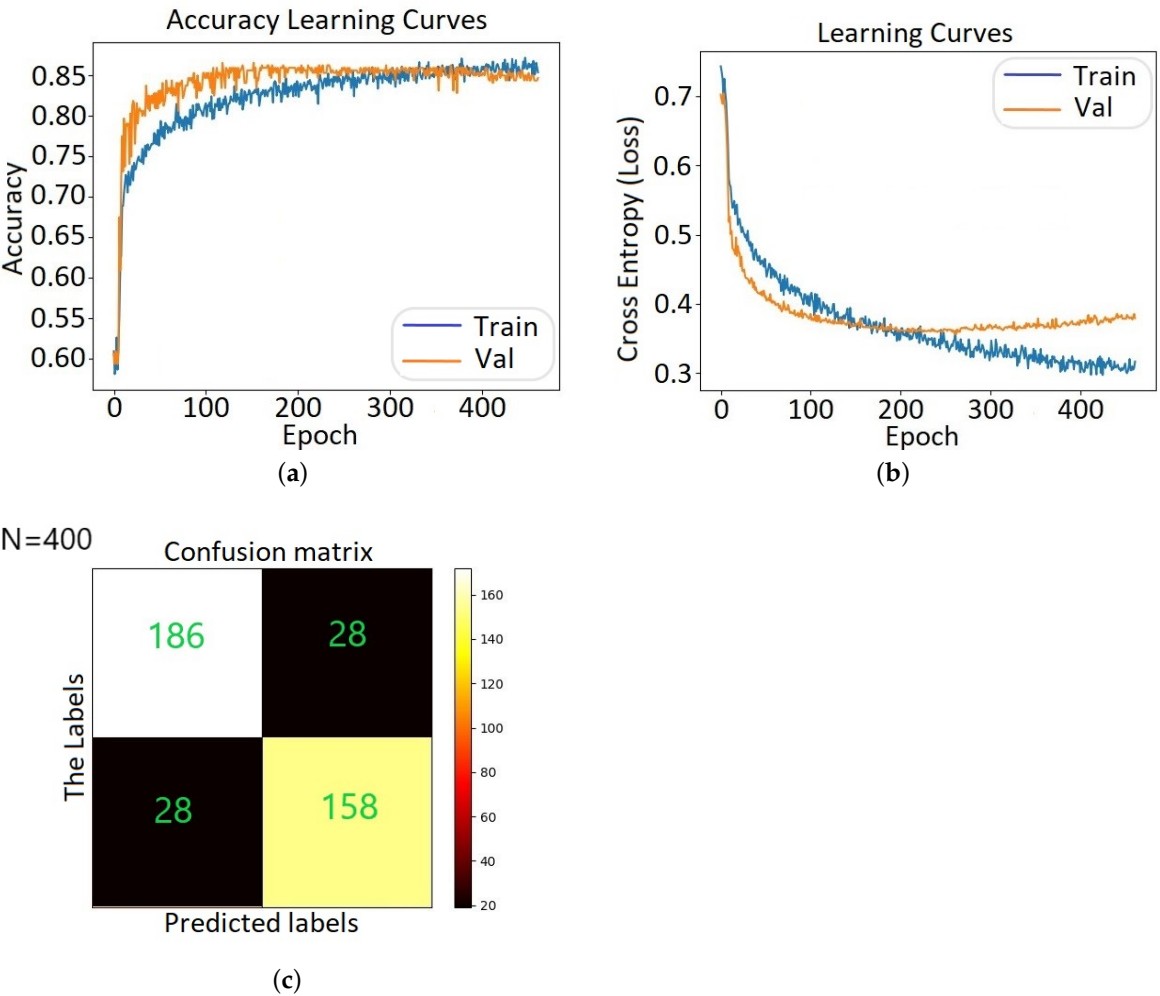

**Figure 10.** Performance curves of the DNN model: (**a**) accuracy; (**b**) cross entropy (loss); (**c**) confusion matrix.

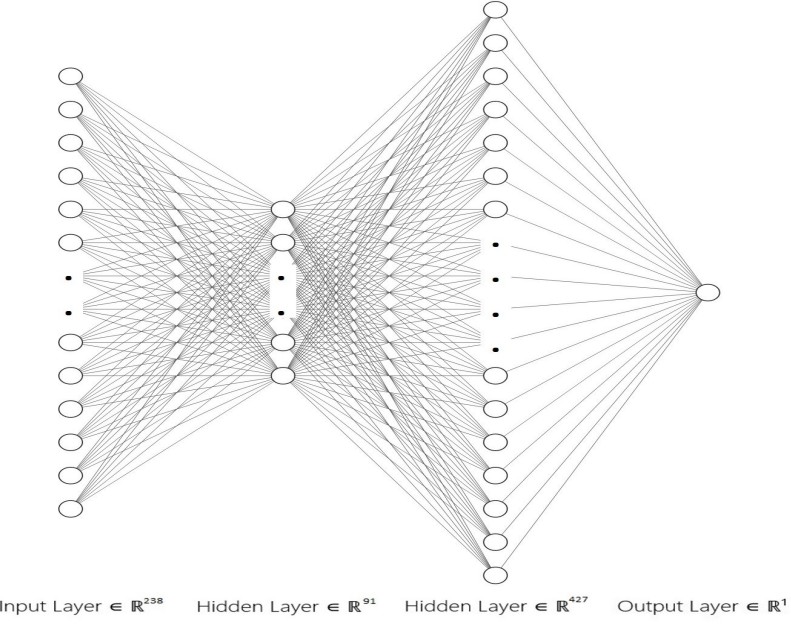

**Figure 11.** DNN architecture of the best performing model. It comprises four layers with 238 (input), 91 (hidden), 427 (hidden), and 1 (output) nodes, respectively.

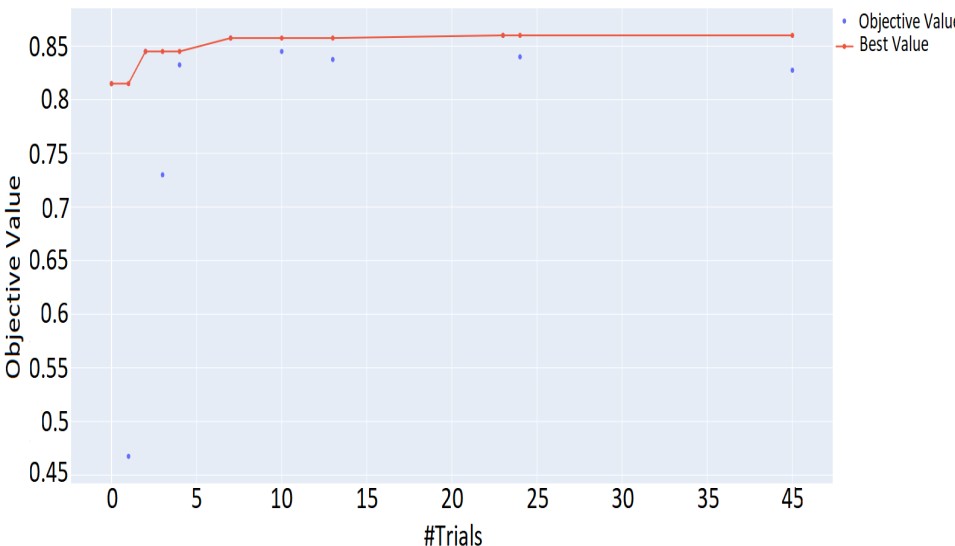

**Figure 12.** Optimization history of the DNN model.

## 6. SHAP Analysis and Features Importance Assessment

Feature importance is generally considered a very useful step when evaluating ML models. During this process, the model is further evaluated and interpreted for deducing which input feature has a positive or negative impact on the model's prediction. For assisting this process, i.e., for interpreting further the DNN model, we relied on the SHAP unified framework [42].

Figures 13 and 14 depict the average impact of the most important input features (mean Shapley values), as well as their contribution (heatmap) to the model's classification output. Precisely, Figure 14 illustrates in descending order the most influencing features in determining the outcome. The high and low values of each input feature are shown in red and blue font, respectively. For example, in the same figure, it is perceived that the com.google.android.c1dm.intent.RECEIVE is the most influencing feature in determining the outcome. Putting it another way, the high values for this feature decrease the possibility for a given application to be malware, while low values augment it.

Figure 13 depicts in descending order for each input feature the assessment of the average magnitude impact on the model's output. Note that both the aforementioned figures show only the features that contribute to the model's prediction, discarding the rest of them. For instance, by observing both Figures 13 and 14, it is obvious that the Intent.RECEIVE feature has a major impact, over 0.08 (SHAP value), to the output of the model. Additionally, low values of this feature positively affect the output of the model, meaning that the model makes more accurate predictions at about 0.1 (SHAP value). Similarly, high values of the same feature negatively affect the output, meaning that the model makes wrong predictions at almost –0.3 (SHAP value). On the other hand, the category.DEFAULT feature impacts the model output at about 0.06. The low values of this feature contribute negatively to the mode's output, whereas high values have a positive impact, at about 0.1. Finally, it is noteworthy that only 7 out of 20 features are Android permissions, while the rest are Intents.

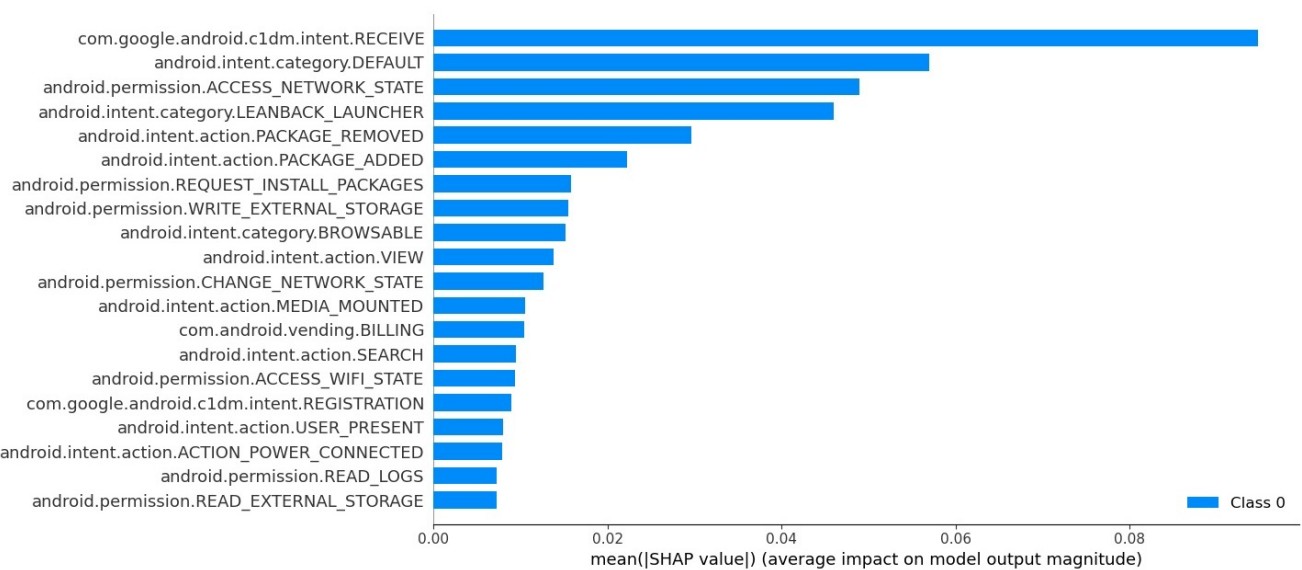

**Figure 13.** Features importance for the DNN model.

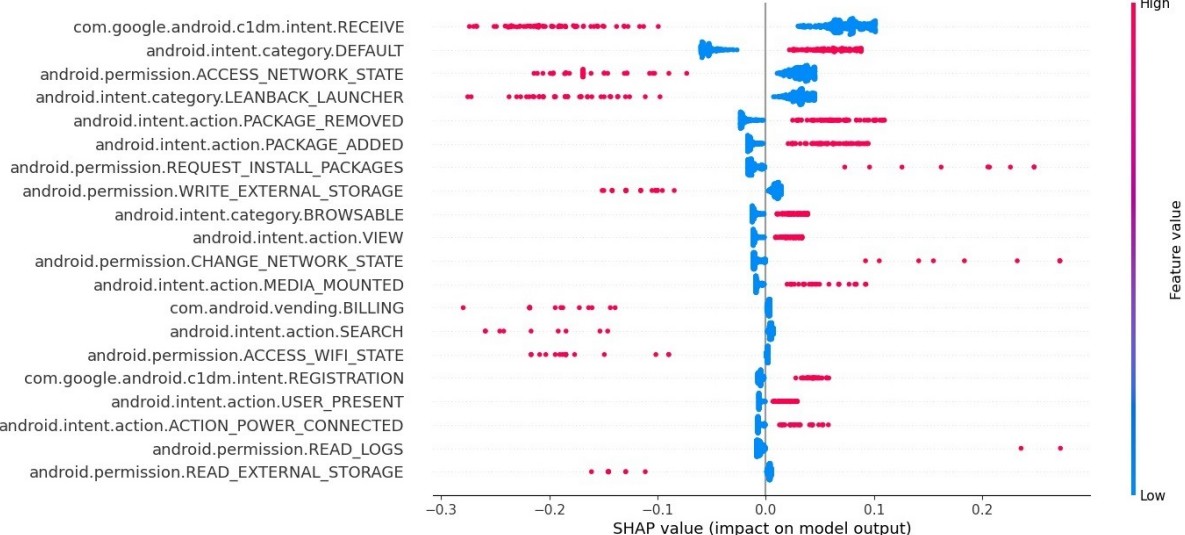

**Figure 14.** Heatmap of the input features for the DNN model.

## 7. Conclusions

The current paper sheds more light on the capability of using both shallow and deep ML techniques for predicting malware in the Android platform. This was achieved by exploring, optimizing, and evaluating the performance of 28 different ML algorithms, including a DNN model.

The optimization process considered hyperparameter analysis by means of the Optuna framework. The most accurate model was found to be the DNN one, succeeding a prediction accuracy of 86%. The evaluation process also included the calculation of diverse performance metrics, such as the balanced accuracy, the F1-score, and the ROC-AUC, which were found to be 82% for the (superior vis-à-vis shallow classifiers) DNN model. Overall, the proposed structure of the DNN model consists of four layers (two hidden), using the Adamax optimizer, as well as the Binary Cross-Entropy (loss), and the Softsign activation functions.

Furthermore, the DNN model was also interpreted for extracting the most important input features that contribute positively or negatively to the final prediction of the model.

This was achieved by means of the SHAP unified framework. However, more input data are needed in order to safely argue about the performance of the DNN model. Finally, as part of future work, we intend to integrate the DNN model into a malware detection application for Android users.

**Author Contributions:** Conceptualization, F.G. and V.K.; Data curation, F.G., V.K. and G.K.; Formal analysis, F.G., V.K. and G.K.; Investigation, F.G. and V.K.; Methodology, F.G., V.K. and G.K.; Resources, V.K. and G.K.; Software, F.G. and V.K.; Supervision, F.G., V.K. and G.K.; Validation, F.G., V.K. and G.K.; Visualization, F.G., V.K. and G.K.; Writing—original draft, F.G., V.K. and G.K.; Writing—review and editing, F.G., V.K. and G.K. All authors have read and agreed to the published version of the manuscript.

**Funding:** This research received no external funding.

**Institutional Review Board Statement:** Not applicable.

**Informed Consent Statement:** Not applicable.

**Data Availability Statement:** The Python scripts as well as the list of malicious and benign samples (applications) used in the experiments can be found in a publicly accessible GitHub repository at https://github.com/billkoul/AndroidMalwareDL (accessed on 17 December 2022).

**Conflicts of Interest:** The authors declare no conflict of interest.

## Abbreviations

The following abbreviations are used in this manuscript:

| | |
|---|---|
| AI | Artificial Intelligent |
| AMD | Android Malware Dataset |
| APK | Android Package Kit |
| AUC-ROC | Area Under The Curve-Receiver Operating Characteristics |
| CNN | Convolutional Neural Networks |
| DNN | Deep Neural Network |
| DT | Decision Tree |
| FFN | Feed-Forward network |
| FN | False Negative |
| FPR | False Positive Rate |
| FP | False Positive |
| LR | Logistic Regression |
| LSTM | Long Short-Term Memory |
| ML | Machine Learning |
| NN | Neural Networks |
| RNN | Recurrent Neural Network |
| SHAP | Shapley Additive Explanations |
| SVM | Support Vector Machine |
| TN | True Negative |
| TP | True Positive |
| VM | Virtual Machine |

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
