# Peer review of "A Closer Look at Machine Learning Effectiveness in Android Malware Detection"

_information, doi:10.3390/info14010002_

Round 1

Reviewer 1 Report

The paper presents a novel method to detect Android malware based on Artificial intelligence

The state of the art is well organized and based on 39 reference titles, most of them recent or very recent

There are several small issues that have to be solved, as follows:

- related to table 1, as it is – it does not say much; please explain (either in the text, or detail in the table) what means (+) (-) (+8) regarding the features mentioned

- regarding to Figure 3. Optimization history of the XGboost classifier – please explain in the text what is the significance of the blue dots and the red line.

- please increase the dimensions of the numbers and labels in Figure 5. Hyperparameters parallel coordination view for the XGboost classifier – it is really hard to read them even increasing to 150%

- in eqn (7) define the variables in the text – who are L, x, h^(k)(b). In the text use the same equation font-face, otherwise it is confusing.

- for Figure 8. Hyperparameters parallel coordination schema of the DNN model, again, please increase the dimension of the text.

- for Figure 14. Heatmap of the input features for the DNN model – please explain what is represented with blue and what with red.

Author Response

Please find attached a detailed response to your comments.

Thank you.

Reviewer 2 Report

1. For Table 1: Provide a bit more description in the caption about what each symbol represents: +, -, and numbers in parenthesis. 

2. Explain why only 'permission' and 'intent' were used for analysis. 

3. Are the two features mentioned above enough to provide a dependable solution?

4. Figure 1 font size is too small. Replot the graphs with larger font. 

5. Put numbers in the confusion matrix blocks. 

6. Section 5.1 is irrelevant. Everyone understands activation functions. There is no need to do a comparative analysis here. 

7. Unfortunately, I am falling short of understanding the bigger picture here. What will this be used for? No mention of transfer learning or model sharing? What is the outcome of this research other than just feature engineering and deep learning? 

8. Where is the code or the API for researchers? 

Author Response

(The authors gave the same response as above.)

Round 2

Reviewer 2 Report

1. Most concerns have been addressed. However, GitHub access to resources have still not been added for reference. This should be addressed before publication. 

2. Figure DPI maybe satisfactory but the text inside figurers are small. See Figure 7 and 9. Replot the graphs in a higher resolution and increase font size. There maybe more so ensure proper modifications in the manuscript.

3. Figure 10 needs more explanation. A similar trend of figures with vague captions are seen in the paper. This should be addressed prior to publication. 

Round 3

Reviewer 2 Report

1.  In this era, majority of the work done utilizes existing libraries in Python. The purpose of providing GitHub access is to provide other researchers an opportunity to look at the workflow of your proposed model. This increases the visibility of your paper and also allows other researchers to build upon your work and not reinvent the wheel. It is advisable to share your .py files or your Jupyter notebook for reference. 

2. Changing orientation from portrait to landscape did not solve the issue that your font size for images are extremely small. If you are using matplotlib, then replotting the images by specifying a larger font size should be pretty straightforward.

3. Figure 10 was an example of vague captions that exist in several figures. It is also advisable to add some context to captions rather than saying "Image of ..."

To summarize, these concerns were partially met or not met at all. But since they don't have to do with content, I feel its okay to accept. 

Author Response

Please find attached the point-to-point response to the reviewer's comments.
